# Design of Amine-Modified Zr–Mg Mixed Oxide Aerogel Nanoarchitectonics with Dual Lewis Acidic and Basic Sites for CO_2_/Propylene Oxide Cycloaddition Reactions

**DOI:** 10.3390/nano12193442

**Published:** 2022-10-01

**Authors:** Yi-Feng Lin, Yu-Rou Lai, Hsiang-Ling Sung, Tsair-Wang Chung, Kun-Yi Andrew Lin

**Affiliations:** 1Department of Chemical Engineering and Research Center for Circular Economy, Chung Yuan Christian University, Chungli District, Taoyuan 32023, Taiwan; 2R&D Center for Membrane Technology, Chung Yuan Christian University, Chungli District, Taoyuan 32023, Taiwan; 3Department of Environmental Engineering & Innovation and Development Center of Sustainable Agriculture, National Chung Hsing University, South District, Taichung 402, Taiwan

**Keywords:** carbon dioxide, Zr–Mg mixed oxide aerogel, cycloaddition reaction, Lewis acid, Lewis base

## Abstract

The utilization of CO_2_ attracts much research attention because of global warming. The CO_2_/epoxide cycloaddition reaction is one technique of CO_2_ utilization. However, homogeneous catalysts with both Lewis acidic and basic and toxic solvents, such as DMF, are needed in the CO_2_/epoxide cycloaddition reaction. As a result, this study focuses on the development of heterogeneous catalysts with both Lewis acidic and basic sites for the CO_2_ utilization of the CO_2_/epoxide cycloaddition reactions without the addition of a DMF toxic solvent. For the first time, the Zr–Mg mixed oxide aerogels with Lewis acidic and basic sites are synthesized for the CO_2_/propylene oxide (PO) cycloaddition reactions. To further increase the basic sites, 3-Aminopropyl trimethoxysilane (APTMS) with -NH_2_ functional group is successfully grafted on the Zr–Mg mixed oxide aerogels. The results indicate that the highest yield of propylene carbonate (PC) is 93.1% using the as-developed APTMS-modified Zr–Mg mixed oxide aerogels. The as-prepared APTMS-modified Zr–Mg mixed oxide aerogels are great potential in industrial plants for CO_2_ reduction in the future.

## 1. Introduction

Recently, the amount of CO_2_ in the atmosphere has significantly increased since the industrial revolution due to the combustion of huge amounts of fossil fuels. Carbon dioxide (CO_2_) as the major greenhouse gas has raised a great deal of attention due to its effect on global warming and climate change [1,2,3]. Without action toward stemming CO_2_ emissions, global warming could pose a massive threat to our life. In order to alleviate the potential threat, there are increasing demands to find new strategies to reduce carbon dioxide and prevent future global problems. In the past, carbon dioxide capture and storage were considered essential strategies for CO_2_ emission reduction targets [4,5,6,7,8]. However, the storage technology poses security concerns and also consumes additional energy during the transportation process. As a result, CO_2_ utilization has drawn much research interest in recent years. Carbon dioxide is not only reduced but also can serve as a nontoxic, low-cost, and renewable C_1_ raw material, replacing other toxic organic solvents and chemical materials for synthesizing fine chemicals and fuels, such as urea, salicylic acid, cyclic carbonates, and polypropylene carbonate [9,10,11]. Out of much CO_2_ utilization, the most promising reaction schemes of CO_2_ currently being studied is the coupling of CO_2_ and epoxides for five-member cyclic carbonates synthesis (Figure 1) [12,13,14,15,16]. In addition, cyclic carbonates are widely applied as electrolytes in lithium-ion batteries, polar aprotic solvents used extensively as intermediates in the production of fine chemicals, such as plastics, and pharmaceutical materials [17,18,19]. Furthermore, cyclic carbonates are important precursors for polymerization. These five-member cyclic carbonates were conventionally manufactured via a corrosive, poisonous, and hazardous route involving glycol and phosgene [20]. So, we shifted to production based on the cycloaddition of CO_2_ to the epoxide, which is less toxic and easier to handle nowadays. Polycarbonates are highly valuable polymeric materials (engineering plastics) that possess outstanding properties, including impact resistance, electrical insulation, and optical properties [21].

However, it is not easy to activate such a thermodynamically stable material (CO_2_). Therefore, catalysts play a significant role in the transformation of carbon dioxide into cyclic carbonates. Thus far, various heterogeneous and homogeneous catalysts have been developed for CO_2_/epoxide cycloaddition reactions. Quaternary ammonium and phosphonium salts [22,23], ionic liquids [24,25,26,27], transition metal complexes [28,29], salen complex [30,31], and alkali metal salts [32,33] have been extensively studied as homogeneous catalysts. Normally, homogeneous catalysts show higher catalytic activity than heterogeneous catalysts in the presence of cocatalysts, but the separation and purification of the homogeneous catalysts from the products are greatly difficult, restricting their wide applications. Hence, the research in CO_2_ cycloaddition reactions has been mostly devoted to the development of highly efficient heterogeneous catalysts, such as metal oxide [34] and metal organic frameworks [35,36,37,38,39,40,41,42], which are easily separated by centrifugation or filtration and can be reused during the CO_2_ cycloaddition reactions.

Several types of heterogeneous catalysts, including metal oxides, have been applied for the CO_2_ cycloaddition reaction. The first example of polycarbonate synthesis via CO_2_ cycloaddition reaction was discovered by Yano et al. [43]. They showed that MgO could catalyze CO_2_ with propylene oxide (PO) to obtain cyclic carbonates. For this reaction, the MgO catalyst gave a poor product yield because of low activity. Bhanage et al. [44] employed several metal oxide catalysts, such as MgO, CaO, and ZrO_2_, for the synthesis of propylene carbonate (PC). The yield and selectivity of the as-synthesized carbonates using different catalysts are also compared. The effects of the Lewis acid/base sites of the catalysts on the cycloaddition reaction are also discussed in detail. Among the metal oxides, ZrO_2_ with acidic catalytic sites gives only 21% conversion and low selectivity of 50%. Shortly later, ternary metal oxides were proposed by Yamaguchi et al. [45]. They found that the Mg–Al mixed oxides obtained by the calcination of the hydrotalcites showed high performance for CO_2_ cycloaddition reactions to form the corresponding five-member cyclic carbonates. The results showed that the optimization condition (88% of product yield) for CO_2_ cycloaddition reaction had the Mg/Al ratio of five calcine at 400 °C in the presence of toxic DMF cosolvent. Based on the catalyst characterization results, the O and Al atoms involved in Mg–O–Al bonds act as basic and acidic sites, which can cooperatively activate epoxide molecules. In addition, doping with Mg will attract more CO_2_ molecules to react with the epoxide. As a result, the heterogeneous catalysts which possess acid–base bifunctional properties are very crucial for CO_2_ cycloaddition reactions.

Unfortunately, conventional heterogeneous catalysts such as nanoparticles are sometimes less active because of aggregation, resulting in a small specific surface area. Therefore, nanoporous materials have attracted much research attention owing to their high specific surface area. Aerogels containing approximately 99% volume of air with large surface areas were first discovered by Kistler in the 1930s [46]. Aerogels are a class of porous materials composed of three-dimensional network structures, having unique physicochemical properties, such as high porosity, large specific surface area, and known to possess lower density properties, which allow potential applications in the field of separation and catalysis [47,48,49]. Apart from the conventional aerogels, several types of aerogel have recently been developed based on different types of precursors, such as inorganic, organic, and carbon. The usual route for aerogel preparation is the sol–gel technique, generally accompanied by supercritical drying technology to afford highly porous structures [50,51,52]. This allows the liquid to be slowly dried off without causing the solid network in the gel to collapse from capillary action. Another method (the epoxide-assisted sol–gel route) for the synthesis of gels was first proposed by Gash et al. [53,54,55]. This method can adjust the pH value of the solution and also promote the hydrolysis and condensation reactions of the hydrate metal salt precursor in water or another solvent to form a gel. This method has been proven successful and environmentally friendly for the synthesis of metal oxide aerogels. Zirconia (ZrO_2_) has been widely used as an advanced material in various applications, including electronics, catalysis, and high-temperature structural engineering because of its superior acidic catalytic characteristics, redox properties, electrical properties, and high thermal and chemical stability [56,57].

In this work, for the first time, we use zirconium oxychloride octahydrate and magnesium nitrate hexahydrate as precursors, epoxide as the irreversible proton scavenger and gelation agent in combination with supercritical drying approaches to prepare Zr–Mg mixed oxide aerogels with high specific surface areas and porosities. The as-prepared Zr–Mg mixed oxide aerogels with acid–base functionalities have great potential as an excellent heterogeneous catalyst for CO_2_/epoxide cycloaddition reactions. Furthermore, we also discuss the effect of different reaction parameters, including temperature, CO_2_ pressure, and reaction time, on the catalytic performance of the CO_2_/epoxide cycloaddition reactions. We plot a schematic diagram (Figure 2) and hope that the as-developed Zr–Mg mixed oxide aerogels can improve not only the performance of catalytic efficiency in the absence of any cocatalysts and solvents but also achieve the vision of CO_2_ reduction and the perspectives of green chemistry in industrial plants.

## 2. Experimental

### 2.1. Synthesis of Zr–Mg Mixed Oxide Aerogels

Zr–Mg mixed oxide precursor gels were synthesized by the epoxide-assisted sol–gel route. Zirconium (IV) oxychloride octahydrate (ZrOCl_2_·8H_2_O), magnesium nitrate hexahydrate (Mg(NO_3_)_2_·6H_2_O), ethanol (CH_3_CH_2_OH), propylene oxide (PO), and HNO_3_ (65 wt%) were used as reactants for the preparation of Zr–Mg mixed oxide aerogels, as shown in Figure 3. According to different Zr/Mg molar ratios, ZrOCl_2_·8H_2_O, Mg(NO_3_)_2_·6H_2_O, and HNO_3_ were added into ethanol solution. The above solution was stirred until the particles were dissolved. After that, PO was added to the resulting solution. The total molar ratio of the precursor, HNO_3,_ and PO was kept at a constant ratio of 1:1:16; the precursor contained ZrOCl_2_·8H_2_O and Mg(NO_3_)_2_·6H_2_O. The solution was shaken for a while and sat for 24 h to form a gel. For the aging process, the gel was washed with ethanol every 24 h for 2 days to ensure the solvents in the gel were replaced by ethanol. Then the gel was dried in an autoclave with supercritical carbon dioxide solvents. The pressure was set at 1200 psi, and the temperature was set at 75 °C. After 4 h drying, the Zr–Mg mixed oxide aerogel was successfully obtained. For the surface modification of Zr–Mg mixed oxide aerogels, the aged gels were immersed into 15 wt% 3-Aminopropyl trimethoxysilane (APTMS)/EtOH solution and refreshed the 15 wt% APTMS/EtOH solution every 24 h for 1, 2, 3, 4, 5 times.

### 2.2. Carbon Dioxide/Epoxide Cycloaddition Catalytic Reactions

The amount of 0.2 g of the as-prepared catalyst, 1.16 g of PO, and a stirring bar were added in a high-pressure reactor. After all screws were locked, the high-pressure reactor was pressured to 10 kgW/cm^2^ with CO_2_. Until the temperature of the high-pressure reactor was heated to the desired value (100, 120, 150, 180, and 200 °C in this case) using the heating tapes and controlled the magnetic stirrer at 200 rpm for the cycloaddition reaction. After the reaction, the high-pressure reactor was put into the ice bath to cool down. The solid–liquid mixture with the liquid products of propylene carbonate (PC) and the powder catalysts of the APTMS-modified Zr–Mg–O aerogels are observed in the reactor. Both can be separated by the centrifugation process with the rotating speed of 3000 rpm for 3 min. The liquid products are further detected by Nuclear Magnetic Resonance Spectroscopy (NMR) to measure the conversion, selectivity, and yield of the CO_2_/PO cycloaddition reaction.

### 2.3. Characterization

The structures and morphologies of aerogel samples with different Zr/Mg molar ratios were investigated using field emission scanning electron microscopy (FESEM, S-4800, Hitachi Inc., Chiyoda City, Japan). The elemental composition of the as-prepared samples was performed by an energy dispersive spectrometer (EDX, Horiba Model 7021H, Hitachi Inc., Chiyoda City, Japan), an accessory of FESEM. The pore size distribution and the specific surface area of the as-prepared aerogels were measured using nitrogen adsorption/desorption isotherms (BET, ASAP 2020, Micromeritics Inc., Norcross, GA, USA), following the Brunauer–Emmett–Teller (BET) theory. The thermal stability of the as-prepared samples was studied by thermogravimetric analysis (TG analysis, TA-Q50, DuPont Inc., Hayward, CA, USA). Furthermore, the functional groups of the samples with different times of APTMS modification were studied by Fourier transform infrared spectroscopy (FTIR, Tensor 27, Bruker Inc., Billerica, MA, USA). The conversion, selectivity, and yield of the CO_2_/PO cycloaddition reaction were measured by NMR (AVIII-400(B662), Bruker Inc., Billerica, MA, USA).

## 3. Results and Discussion

### 3.1. The Synthesis of Zr–Mg Mixed Oxide Aerogels

The surface morphologies of the as-prepared Zr–Mg mixed oxide aerogels using the addition epoxide method were first investigated by FESEM. Figure 1 shows the FESEM images of the as-prepared Zr–Mg mixed oxide aerogels with 10:0 (Figure 1a), 9:1 (Figure 1b), 8:2 (Figure 1c), 7:3 (Figure 1d), and 6:4 (Figure 1e) molar ratio of Zr to Mg, respectively. The FESEM pictures show that the large clusters were observed for the as-prepared Zr–Mg mixed oxide aerogels with a 6:4 molar ratio of Zr to Mg, which indicates the aggregation of the particles as the Mg molar proportion increased. The Zr, Mg, and O EDS elemental mapping images of the Zr–Mg mixed oxide aerogels with a 7:3 molar ratio of Zr to Mg are shown in Figure 2. The Zr, Mg, and O atoms were dispersive very well from the EDS elemental images, implying Zr, Mg, and O atoms are distributed uniformly in the as-prepared Zr–Mg mixed oxide aerogels.

The specific surface areas and pore size distribution of the as-prepared Zr–Mg mixed oxide aerogels with different Zr/Mg molar ratios were measured using nitrogen adsorption/desorption isotherms, as shown in Figure 3 and Figure 4. Type IV N_2_ adsorption/desorption isotherms were observed for all Zr–Mg mixed oxide aerogel samples, indicating the existence of mesopores in the mixed oxide aerogels. This is in good agreement with the average pore size between 2–5 nm for all Zr–Mg mixed oxide aerogel samples, as shown in Table 1. The pore size distributions of the Zr–Mg mixed oxide aerogel samples are narrow and concentrated between 2–10 nm, as shown in Figure 4. The specific surface areas of Zr–Mg mixed oxide aerogel samples shown in Table 1 are 465 (Zr/Mg of 10/0), 283 (Zr/Mg of 9/1), 365 (Zr/Mg of 8/2), 371 (Zr/Mg of 7/3), and 261 (Zr/Mg of 6/4) m^2^/g, respectively. The specific surface areas of the Zr–Mg mixed oxide aerogels decrease from 465 m^2^/g to 261 m^2^/g when the Mg molar ratios of the aerogels increase. This is probably because the addition of Mg destroys the gel structures and causes the pores to collapse. Although the specific surface area of Zr–Mg mixed oxide aerogels is less than that of the pristine ZrO_2_ aerogels, the specific surface area of Zr–Mg mixed oxide aerogels with Zr/Mg molar ratio of 7/3 still reaches the value of 371 m^2^/g, which is much higher than that of other composite metal oxides.

Since the carbon dioxide cycloaddition reaction requires a heating process to energize the environment, thermogravimetric analysis is used to analyze the thermal stability of the Zr–Mg mixed oxide aerogels with Zr/Mg molar ratio of 7/3, which is the catalyst with highest PC yield of CO_2_/PO cycloaddition reaction. Figure 5 is the picture of TG analysis of the Zr–Mg mixed oxide aerogels with a Zr/Mg molar ratio of 7/3. It shows that the weight loss before 200 °C is inferred to be the surface water (-OH functional group) removal of the aerogels. This phenomenon may cause the aerogel structures to collapse and affect the activity of the catalysts during the cycloaddition reaction, resulting in poor reactivity of the catalyst.

The bonding configurations of Zr (3d) (Figure 6a), O (1s) (Figure 6b), and Mg (1s) (Figure 6c) atoms in the pristine ZrO_2_ aerogel and Zr–Mg mixed oxide aerogels with Zr/Mg molar ratio of 7/3 were examined using X-ray photoelectron spectroscopy (XPS), as shown in Figure 6. Zr 3d curves (Figure 6a) are fitted to two peaks corresponding to 3d_5/2_ and 3d_3/2_ lines, respectively. Zr 3d_5/2_ and Zr 3d_3/2_ peaks are located at 182.1 eV and 184.5 eV for the pristine ZrO_2_ aerogel, as shown in Figure 6(i) of Figure 6a. The distance between two satellites of Zr 3d_5/2_ and Zr 3d_3/2_ for the pristine ZrO_2_ aerogel is 2.4 eV, which is in good agreement with the reference ZrO_2_ satellite peaks of Zr 3d_5/2_ and Zr 3d_3/2_. This means ZrO_2_ aerogels are successfully synthesized in this work. On the other hand, the position of Zr (3d) peaks in the Zr–Mg mixed oxide aerogels with Zr/Mg molar ratio of 7/3 (Figure 6(ii) of Figure 6a) are chemically shifted toward larger binding energy values of 183.6 eV for Zr 3d_5/2_ peak and 185.9 eV for Zr 3d_3/2_ peak compared with the peaks in the pristine ZrO_2_ aerogels. This indicates that the chemical properties of Zr^4+^ species in the as-prepared Zr–Mg mixed oxide aerogels with a Zr/Mg molar ratio of 7/3 are different from those in pure ZrO_2_ aerogels. As for the O 1s peak (Figure 6b), ZrO_2_ aerogel (Figure 6(i) of Figure 6b) shows the peak centered at a binding energy of 530.9 eV, which corresponds well with the reference ZrO_2_ satellite peak of O 1s. However, the position for O 1s in the as-prepared Zr–Mg mixed oxide aerogels with Zr/Mg molar ratio of 7/3 is chemically shifted to a large binding energy value of 532.3 eV. The results of the Zr 3d and O 1s curves for the ZrO_2_ and Zr–Mg mixed oxide aerogels indicate that there is another atom to bond with Zr–O bonds to form the solid solution. Furthermore, the Mg 1s peak (Figure 6c) shifted to a higher binding energy value of 1305.6 eV compared with the reference Mg 1s peak of Mg and MgO. The result indicates that Mg is successfully doped into the ZrO_2_ aerogel for the Zr–Mg mixed oxide aerogels with a Zr/Mg molar ratio of 7/3.

Figure 7 shows the PC yields of the CO_2_/PO cycloaddition reactions with different Zr/Mg molar ratios of Zr–Mg mixed oxide aerogels (10/0, 9/1, 8/2, 7/3, and 6/4) under reaction temperature of 150 °C, the pressure of 10 kgW/cm^2^, and reaction time of 15 h. More clearly, the yield of PC significantly enhanced with the increase in Mg amount for the as-prepared Zr–Mg mixed oxide aerogels. The detailed data of the PO conversion, selectivity, and PC yield are given in Table 2. Zr–Mg mixed oxide aerogels with Zr/Mg molar ratio of 10/0 and the pristine ZrO_2_ aerogel have large PO conversion of 91% and low PC selectivity of 9.4% because of rich Lewis acidic and fewer Lewis basic sites of the pristine ZrO_2_ aerogels. When the amount of Mg is increased for the Zr–Mg mixed oxide aerogels, the PO conversions are slightly increased; however, the PC selectivity is greatly enhanced from 9.4 to approximately 65%. This is because the Lewis basic sites for the Zr–Mg mixed oxide aerogels are successfully increased with the addition of Mg into the aerogels. It can be found that the molar ratio with the Zr/Mg molar ratio of 7/3 gives the highest PC yield (63.6%), but PC yield decreases even if the basic species of Mg amount is increased to the Zr/Mg molar ratio of 6/4. This is because the specific surface area for the Zr–Mg mixed oxide aerogels with a Zr/Mg molar ratio of 6/4 (260.77 m^2^/g) is lower than that with Zr/Mg molar ratio of 7/3 (370.67 m^2^/g).

### 3.2. Mg–Zr Mixed Oxide Aerogels with Amino Functional Group Modification

As mentioned, the PC yield rises with the increase in the Mg incorporation into ZrO_2_ aerogels to enhance the Lewis basic sites. However, the obtained PC yield of 63.6% using Zr–Mg mixed oxide aerogel catalysts with the Zr/Mg molar ratio of 7/3 is still not large enough. In order to increase the Lewis basic sites of the as-prepared aerogel surfaces, APTMS with an amino functional group is grafted on the surface of Zr–Mg mixed oxide aerogels with the Zr/Mg molar ratio of 7/3 for the enhancement of Lewis basic sites of the aerogel surfaces. The relevant catalytic performance using APTMS-modified Zr–Mg mixed oxide aerogels with the Zr/Mg molar ratio of 7/3 will be investigated and discussed later.

The EDS elemental mapping of APTMS-modified Zr–Mg mixed oxide aerogels is measured as shown in Figure 8. Figure 8a–d is the EDS mapping image of Zr (Figure 8a), Mg (Figure 8b), O (Figure 8c), and N (Figure 8d) elements, respectively. We can clearly observe that N signals were well-distributed on the surface of APTMS-modified Zr–Mg mixed oxide aerogels, indicating APTMS agents are successfully grafted on the aerogel surface. To further prove the APTMS successful grafting, FT-IR spectra of the APTMS-modified Zr–Mg mixed oxide aerogels with zero to five times modifications were shown in Figure 9. The peak intensities of the -OH functional group (3250–3500 cm^−1^) decrease with increased APTMS modifications. In contrast, the peak intensities of the -NH_2_ functional group show the opposite trend, decreasing with the increased APTMS modifications. This indicates the -Si-(O-CH_3_)_3_ group of APTMS agents are bonded to the Mg-O-Zr-OH group of the Zr–Mg mixed oxide aerogels to form CH_3_OH and -Si-O-Zr-O-Mg bonds, leading to the decrease in the intensities of -OH functional groups. It also proves that the APTMS agents are successfully modified on the aerogel surfaces. Figure 10 is the TG analysis of the APTMS-modified Zr–Mg mixed oxide aerogels. It shows that the APTMS-modified Zr–Mg mixed oxide aerogel has better thermal stability than the pristine Zr–Mg mixed oxide aerogel without APTMS modifications between 0 and 250 °C. This proves that the APTMS modification effectively removes the attached surface water of the pristine aerogel surfaces, and APTMS modification is beneficial to the enhancement of the thermal stability of the aerogel catalysts.

Figure 11 and Table 3 show the catalytic results of the CO_2_/PO cycloaddition reactions using Zr–Mg mixed oxide aerogels with a Zr/Mg molar ratio of 7/3 under different APTMS modification times. The PC selectivity rises from 65.2% to 95.2% with the increased APTMS modifications from zero to four times, resulting from the effective enhancement of the Lewis basic sites on the surface of APTMS-modified Zr–Mg mixed oxide aerogels with Zr/Mg molar ratio of 7/3. Consequently, the PC yield with four APTMS modification times also reaches the largest value of 93.1% with CO_2_ conversion of 97.7% and PC selectivity of 95.2%. However, the PC selectivity decreases to 91%, leading to the decline of the PC yield (89.5%) when the APTMS modification increases to five times. This is because larger grafted APTMS agents on the aerogel surface have hindered the surface of Zr–Mg mixed oxide aerogels in their effort to catalyze CO_2_/PO cycloaddition reactions.

Figure 12 and Table 4 exhibit the results of PO conversion, PC selectivity, and PC yield using APTMS-modified Zr–Mg mixed oxide aerogels at the different reaction temperatures of 100, 120, 150, 180, and 200 °C, respectively. PC selectivity and yield slightly enhance with the increased reaction temperature from 100 to 150 °C and reach the largest PC yield of 93.1% at a reaction temperature of 150 °C. However, when the reaction temperature still goes up to 180 and 200 °C, the PC yield and selectivity both go down to 67.2% rather than increase the value. It means there is an optimal reaction temperature (150 °C) for this CO_2_/PO cycloaddition reaction. Kongpanna et al. [58] claimed the enthalpy (ΔH) of this CO_2_/PO cycloaddition reaction is −0.808 kJ/mol, the so-called exothermic reaction, which makes the reaction likely favors low temperature and high pressure to synthesize the main product (PC) instead of the side products (PPO).

The results of PO conversion, PC selectivity, and PC yield using APTMS-modified Zr–Mg mixed oxide aerogels at the different reaction times of 6, 9, 12, 15, and 24 h are shown in Figure 13 and Table 5. The PC yield at the reaction time of 6 h (83.9%) is lower than that of 15 h (93.1%), as shown in Figure 13 and Table 5. It means there is enough effective catalytic interaction between reactants (CO_2_ and PO) and APTMS-modified Zr–Mg mixed oxide aerogel catalyst. However, if the reaction time still increases to 24 h, PC yield decreases rapidly to 40.6% because of the tendency to form PPO by-products. As a result, the optimal reaction time is 15 h to obtain the largest PC yield using APTMS-modified Zr–Mg mixed oxide aerogel catalysts.

The catalytic performance of PC yield using APTMS-modified Zr–Mg mixed oxide aerogels were also compared with other studies, as shown in Table 6. For metal oxide catalysts, the addition of a cocatalyst or solvent is necessary to enhance their efficiencies of cycloaddition reactions. However, the catalytic efficiency using metal oxide catalysts is still quite low. As a result, some literature focus on the preparation of nonsolvent and cocatalyst-free MOF catalysts for cycloaddition reactions. In this study, we successfully designed the APTMS-modified Zr–Mg mixed oxide aerogel nanoarchitectonics for the CO_2_/PO cycloaddition reactions. Table 6 shows that PC yields using APTMS-modified Zr–Mg mixed oxide aerogels in this work reach 93.1%, which is comparable with some MOF catalysts. As a result, the as-prepared APTMS-modified Zr–Mg mixed oxide aerogels show great potential in industrial plants for CO_2_ reduction in the future.

## 4. Conclusions

For the first time, Zr–Mg mixed oxide aerogels were successfully synthesized by doping Mg into ZrO_2_ aerogels via the epoxide-assisted sol–gel route. The as-prepared Zr–Mg mixed oxide aerogels were also used for the CO_2_/PO cycloaddition reactions to investigate the PO conversion, PC selectivity, and PC yields. Compared with pristine ZrO_2_ aerogels, Zr–Mg mixed oxide aerogels can effectively enhance the Lewis basic sites of the aerogel surfaces, leading to an increase in the PC selectivity and PC yields. The highest PC selectivity and PC yields are 65.2% and 63.6% for the Zr–Mg mixed oxide aerogels with a Zr/Mg molar ratio of 7/3. However, the PC selectivity of 65.2% is still not large enough, resulting from the lack of Lewis basic sites of the Zr–Mg mixed oxide aerogels with a Zr/Mg molar ratio of 7/3. To increase the Lewis basic sites of the aerogel surfaces, APTMS with amino functional groups were grafted on the surface of the Zr–Mg mixed oxide aerogels with a Zr/Mg molar ratio of 7/3. The largest PC yield of 93.1% was observed using Zr–Mg mixed oxide aerogels with a Zr/Mg molar ratio of 7/3 under four-time APTMS modification, resulting from the increase in the Lewis basic sites after APTMS modifications. The as-prepared APTMS-modified Zr–Mg mixed oxide aerogels have great potential in future CO_2_/PO cycloaddition reactions.

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
