# Peer review of "Design of Amine-Modified Zr–Mg Mixed Oxide Aerogel Nanoarchitectonics with Dual Lewis Acidic and Basic Sites for CO2/Propylene Oxide Cycloaddition Reactions"

_nanomaterials, 2022, doi:10.3390/nano12193442_

Round 1
Reviewer 1 Report
The authors reported “Synthesis of Zr-Mg Mixed Oxide Aerogels for Carbon Diox-ide/Epoxide Catalytic Reactions”. I believe that the manuscript can be published in Nanomaterials after major revisions.
The authors report an interesting process for the fixation of CO2 in organic substrates by heterogeneous catalysis but it is necessary to specify that the process is truly heterogeneous and thatthe generality of the process..
I'll explain: Scheme 1 shows a generic substituent R on the epoxide, but in the text we speak only of the synthesis of propylene carbonate, therefore with R of methyl nature. I believe that, in order for the manuscript to be considered complete and therefore ready to be accepted, at least two other examples of synthesis of cyclic carbonates in heterogeneous catalysis should be reported. For example, the substituent R of scheme 1 could be a phenyl substituent (therefore of an aromatic nature), and a long-chain or sterically cluttered alkyl substituent.
Could the heterogeneous catalyst be recycled? after the first reaction, could the catalyst already used be added with fresh substrate and solvent and under the same conditions as the parent reaction, under CO2 atmospheres, lead to the formation of cyclic carbonate once again? I believe the authors should prove the recyclability of the catalyst before the manuscript can be accepted.
Furthermore, "hot" and "cold" filtration tests have been carried out on the catalyst to demonstrate that the catalysis is indeed heterogeneous? It also works at 200 ° C, the catalysis could take place in the homogeneous phase. These are “conditio sine qua non” for the acceptance of the manuscript.
In “Carbon Dioxide/Epoxide Cycloaddition Catalytic Reactions” section the authors should detail the recovery procedure of the obtained product. It is not sufficient to report that the reaction crude is centrifuged. All the steps and the purification step of the cyclic carbonate must be explained.
Reviewer 2 Report
It is a good manuscript with lots of data that are well organized in this manuscript. Therefore, I recommend the publication of this work in nanomaterials after minor revisions. Several revisions to improve the quality are necessary.
C1. The current title is not that impressive to indicate the innovative points of this research. Use of a new conceptual term (such as catalyst nanoarchitectonics; ref:https://doi.org/10.1016/j.mattod.2015.08.021 or https://doi.org/10.1039/D0NH00680G) in the title often works very effective. I would suggest revising the title.
C2. The introduction is rather short and weak. It may include a wider range of backgrounds. For example, some related materials such as porous materials or MOFs can be shortly described with citing recent papers, for example,
-https://doi.org/10.1016/j.mcat.2022.112530
-https://doi.org/10.3390/catal11091061
-https://doi.org/10.1016/j.jece.2021.105113
C3. Addition of the initial figure to explain outline of fabrication methods for the catalyst is recommended for easier understanding.
C4. The production of Scheme 2 involving Chemdraw structures needs to be checked for good visualization.
C5. How about the status of the present catalyst relative to the reported ones? At least, a table to show the relative performance of this study is recommended to provide.
Round 2
Reviewer 1 Report
Accepted in the present form